# Optical Biosensor Based on Porous Silicon and Tamm Plasmon Polariton for Detection of CagA Antigen of *Helicobacter pylori*

**DOI:** 10.3390/s24165153

**Published:** 2024-08-09

**Authors:** Guoguang Rong, Alexey Kavokin, Mohamad Sawan

**Affiliations:** 1CenBRAIN Neurotech Center of Excellence, School of Engineering, Westlake University, Hangzhou 310030, China; rongguoguang@westlake.edu.cn; 2The International Center for Polaritonics, School of Sciences, Westlake University, Hangzhou 310030, China; a.kavokin@westlake.edu.cn

**Keywords:** optical biosensor, Tamm Plasmon Polariton, porous silicon, CagA antigen, *Helicobacter pylori*, high sensitivity, specificity, diagnostics

## Abstract

*Helicobacter pylori* (*H. pylori*) is a common pathogen with a high prevalence of infection in human populations. The diagnosis of *H. pylori* infection is critical for its treatment, eradication, and prognosis. Biosensors have been demonstrated to be powerful for the rapid onsite detection of pathogens, particularly for point-of-care test (POCT) scenarios. In this work, we propose a novel optical biosensor, based on nanomaterial porous silicon (PSi) and photonic surface state Tamm Plasmon Polariton (TPP), for the detection of cytotoxin-associated antigen A (CagA) of *H. pylori* bacterium. We fabricated the PSi TPP biosensor, analyzed its optical characteristics, and demonstrated through experiments, with the sensing of the CagA antigen, that the TPP biosensor has a sensitivity of 100 pm/(ng/mL), a limit of detection of 0.05 ng/mL, and specificity in terms of positive-to-negative ratio that is greater than six. From these performance factors, it can be concluded that the TPP biosensor can serve as an effective tool for the diagnosis of *H. pylori* infection, either in analytical labs or in POCT applications.

## 1. Introduction

*Helicobacter pylori* is a very common pathogenic bacterium. According to statistics, more than half of the world’s population is infected with *Helicobacter pylori* [1]. *Helicobacter pylori* is a spiral shaped bacterium, abbreviated as *H. pylori*, that can infect the human stomach. *H. pylori* exists in various regions of the stomach and duodenum, with strong activity and reproductive ability, posing a great threat to human health. *H. pylori* is mainly transmitted through oral–oral or fecal–oral routes of transmission, causing nausea, acid reflux, belching, upper abdominal discomfort, decreased appetite, and bad breath [2]. The diagnostic of *H. pylori* infection is very important for infection diagnosis, controlling its activity, confirming its eradication, and restoring health [3].

Depending on whether endoscopy is used, diagnostic techniques for *H. pylori* can be classified into two main categories: non-invasive and invasive techniques. Non-invasive techniques include urea breath test [4], stool antigen test [5], serological test [6], and biosensing [7,8,9]. Invasive techniques include endoscopy [10], histology [11], culturing [12], rapid urea test [13], and polymerase chain reaction [14]. We summarize and compare in Table 1 the currently available diagnostic techniques for *H. pylori* reported in academic literature. Also included in Table 1 is the biosensor technique, which is discussed in the following.

Researchers have demonstrated that biosensors are powerful tools for the rapid onsite detection of pathogens, antigens, and biomarkers. Due to their high sensitivity, good specificity, ease of operation, and low cost, biosensors have also been used for the diagnosis of *H. pylori* infection. For example, an electrochemical biosensor was used to detect the DNA of *H. pylori* with a limit of detection (LOD) of 0.06 µg/mL and a dynamic range of 0.72–7.92 µg/mL [7]. In another work, a piezoelectric biosensor has been used to serologically determine *H. pylori* infection through the detection of IgG antibodies with biosensor signal amplification by a secondary antibody binding with IgG [6]. Moreover, an optical biosensor to detect the urease gene of *H. pylori* by a gold-nanoparticle-labeled probe with a LOD of 0.5 nM has also been proposed [8]. In addition, the colorimetric detection of *H. pylori* through natural pH indicators, together with image processing to achieve an LOD of 10 CFU/mL, has been reported [9]. Finally, an amperometric biosensor for the specific detection of DNA of *H. pylori* with a quantification range of 5–20 pmol and detection limit of 6 pmol has been demonstrated [15].

Among these available biosensing techniques, optical biosensors have the advantages of high sensitivity, good specificity, easy operation, fast response, portability, and compatibility with optoelectronic integration. To illustrate the advantages, optical biosensors based on waveguides [16], ring resonators [17], resonant microcavities [18], photonic crystals [19], optic fibers [20], surface plasmon resonance (SPR) [21], localized surface plasmon resonance (LSPR) [22], and surface enhanced Raman spectroscopy (SERS) [23] have been proposed as effective detectors of various kinds of bio species, such as biomarkers, proteins, DNAs, virus, bacterium, and cells. On the other hand, from the material perspective of biosensing technology, porous silicon (PSi) is a versatile nanoscale material for biosensing applications. It has a high surface area for biomolecular immobilization, CMOS-compatible fabrication process for integration with electronics or optoelectronics, and facile biofunctionalization approaches based on the Si/SiO_2_ interface. Porous silicon-based optical devices, such as optical waveguide [16], resonant microcavity [18], photonic crystal [24], and optic fiber [25,26] have all been demonstrated to be efficient biosensing platforms.

We have previously proposed a novel optical biosensor based on the nanophotonic phenomenon of an interfacial optical resonant state-Tamm Plasmon Polariton (TPP) [27,28,29]. Porous silicon was used to construct a distributed Bragg reflector (DBR). Gold thin metal film deposited on top of the porous silicon DBR allows the formation of a TPP resonant device. The gold material serves as a plasmonic or epsilon-near-zero material required to sustain the TPP mode [30,31]. TPP is a phenomenon similar to SPR in that both are electromagnetic surface states at the metal–dielectric interface [32,33,34]. SPR only supports transverse magnetic (TM)-polarized waves, but TPP supports both transverse electric (TE) and TM waves. SPR requires the wavevector boosting by prism or periodic grating, but TPP can be directly excited by propagating waves in the air. Compared with SPR, which is a surface state between metal and dielectric, TPP is a surface state at the interface between metal and dielectric DBR. Therefore, the fabrication of a TPP device is more complicated in terms of processes and materials. The signal interrogation of the TPP optical biosensor is based on reflection spectroscopy. Machine learning techniques can also be employed to process the optical signals to obtain reliable detection results [35]. For applications, we demonstrated the detection of the SARS-CoV-2 virus with the porous silicon TPP optical biosensor. We also demonstrated the detection of the nucleocapsid protein of SARS-CoV-2 virus with the PSi TPP optical biosensor. Note that DBR itself can also serve as an optical biosensor. Upon the binding of analyte, the DBR total reflection band will also shift to a longer wavelength. However, since the DBR has no resonant mode condition, the interactions between biomolecules and light field are weak and the spectral shift will be small. Therefore, the DBR itself will not be a high-sensitivity biosensor. This is the reason why Au thin film is deposited on top of DBR to construct a resonant TPP device. The TPP resonant mode can offer a much higher sensitivity in detecting biomolecules than DBR.

In this work, based on our previous related research on TPP-based biosensors with porous silicon material [27,28,29], we propose that the optical PSi TPP biosensor can be used for the CagA antigen detection to diagnose *H. pylori* infection. The PSi TPP biosensor has been demonstrated to have a high sensitivity and a good specificity. It is also compatible with CMOS (complementary metal oxide semiconductor) fabrication processes, promising low-cost mass production and integration with electronics or optoelectronics. On the other hand, after considering many antigen targets for detection, we found that, among the many virulent factors associated with *H. pylori* bacterium, such as CagA, VacA (Vacuolating cytotoxin), and BabA (blood group antigen binding adhesin), the CagA virulent factor is very important for *H. pylori* diagnosis since it has been reported that CagA-positive patients may be related to more severe gastritis [36], and that they may even indicate a greater tendency towards the development of cancer [37] than CagA-negative patients.

The detection of the CagA antigen with the PSi TPP biosensor is based on the interaction of electromagnetic energy confined by the TPP resonant mode with biomolecules, such as the CagA antigen and antibody. The specific detection of the CagA antigen derives from the specific capturing of the CagA antigen by the CagA antibody physically immobilized on the energy confinement region of the TPP biosensor. By optimizing the Au thin film thickness of the PSi TPP device structure through temporal mode coupling theory, the strongest field confinement of the TPP resonant mode can be obtained at the temporal critical coupling condition [31]. Therefore, the highest sensitivity of the PSi TPP biosensor can be obtained due to the maximized interaction between biomolecules and the electrical field. The immobilization strategy of the specific CagA antibodies also plays an important role in the sensitivity and specificity performances of the PSi TPP biosensor. Based on our previous experiences with SARS-CoV-2 N-protein detection, we also developed an immobilization strategy and protocol applicable for CagA antibodies conjugated with the PSi TPP biosensor. All these works are intended for a highly sensitive and specific PSi TPP biosensor for the detection of the CagA antigen with the objective of *H. pylori* diagnosis.

## 2. Materials and Methods

The porous silicon-based TPP device under study is based on the porous silicon distributed Bragg reflector (DBR). First, a single crystalline silicon wafer (6-inch diameter, Boron doped P-type, 0.01–0.02 Ω∙cm resistivity, <100> crystal orientation, ~700 µm thickness, single- or double-side polished, Ferrotec Shanghai Silicon Wafer, Shanghai, China) was soaked in 5% aqueous hydrofluoric acid (HF) solution for 5 min to remove the native oxide of approximately 20 nm on top of the silicon wafer. Then, the porous silicon DBR was fabricated by the electrochemical anodization of the polished side of the single crystalline silicon wafer in 15% aqueous ethanoic hydrofluoric acid (HF) solution (3:7 *v*/*v* of 50% aqueous HF (J&K Scientific, Beijing, China) and 99% ethanol (Sinopharm Chemical Reagent Co., Ltd., Shanghai, China)). During electrochemical anodization, a platinum wire is in contact with HF and serves as the cathode, while an aluminum plate is in contact with the other side of the silicon wafer (either polished or etched) and serves as the anode. The electrochemical anodization conditions and the resulting structural parameters of porous silicon can be found in Table 2 [28]. The resulting porous silicon material has approximately cylindrical pores with an effective circular cross-section of about 20–30 nm in pore diameter.

After electrochemical anodization, the porous silicon DBR was thermally oxidized under 800 °C in an ambient atmosphere for 30 min to convert Si–H groups into more stable Si–O groups. The oxidation process can also make porous silicon more hydrophilic to facilitate the infiltration of the aqueous solution into the nanoscale pores of porous silicon.

Afterwards, a titanium thin film of approximately 5 nm thickness and a gold thin film of approximately 30 nm thickness were consecutively deposited on the oxidized porous silicon DBR through magnetic controlled sputtering. The titanium thin film served as adhesion layer between the PSi and gold thin film. The gold thin film served as epsilon-near-zero material in the TPP device structure [30]. Then, the 6-inch wafer with the TPP device fabricated on top was cut into 5 mm × 5 mm chips with a dice saw. Each chip was a TPP device that served as a PSi TPP biosensor. For multiple detections, we set up an array of biosensor chips in the format of 12 × N, where there was integer N ∈ [1,8]. The center-to-center distance between chips was 9 mm in both horizontal and vertical direction of the array. The resulting chip density was 12,345 chips/m^2^. The PSi TPP device acted as a biosensor after biofunctionalization with specific antibodies, binding proteins, or aptamers.

Finally, the TPP chip was biofunctionalized through a covalent bond immobilization strategy. In general, the surface of the chip was modified with a polycarboxylation reagent, and then the antibody was immobilized using covalent conjugation method. Table 3 describes the biofunctionalization procedure of the PSi TPP biosensor in more detail. Note that the biosensors are intended for single-use only and the storage conditions are given in row 7 of Table 3.

The optical characterization of the PSi TPP biosensor is based on a reflection spectroscopy, which is a commonly used technique in optics. The optical measurement setup is shown in Figure 1. A white light source provided the incident light. The incident light was guided by six circumferential fibers of a Y-shape fiber bundle and impinged on the PSi TPP biosensor surface with a circular shaped beam spot of approximately 1 mm in diameter. The light reflected from the PSi TPP biosensor surface was collected by the central fiber of the Y-shape fiber bundle and then guided towards the spectrometer of pm-resolution for spectral analysis. Visible light of 400–800 nm in wavelength was used to interrogate the reflection spectrum of the PSi TPP biosensor, which is light intensity as a function of wavelength $I_{Original}(\lambda )$.

For the correction of the light source emission spectrum, a silver mirror was used as a reference to obtain $I_{Silver}(\lambda )$. For a correction of background noise due to ambient lighting, the light source was turned off and the reflection spectrum was collected from the PSi TPP biosensor, which is denoted as $I_{BK}(\lambda )$. Then, the actual reflection spectrum of the PSi TPP biosensor after the light source and background corrections is:(1)$$I_{Correct}(\lambda )=\frac{I_{Original}\left(\lambda \right)-I_{BK}\left(\lambda \right)}{{I_{Silver}\left(\lambda \right)-I}_{BK}\left(\lambda \right)}$$

The calculation shown in Equation (1) was carried out automatically by spectral analysis software during each spectrum measurement. The working principles of the PSi TPP biosensor are as follows. Biomolecules binding inside the nanopores of the first porous silicon layer can interact with the strongly confined electrical field of the TPP resonant mode. This electrical field confinement is due to the coupling of incident light, an electromagnetic wave with an energy inversely proportional to the TPP resonant wavelength, into the TPP resonant mode [31]. This interaction between biomolecules and the electrical field results in the perturbation of the electrical field distribution pattern of the TPP resonance mode and will manifest itself as a shift of the TPP resonance wavelength towards longer wavelength in its reflection spectrum [27].

The shift to a longer wavelength is also called “redshift” due to the longer wavelength of red light in the visible spectrum. Therefore, the signal processing of the TPP biosensor is based on detecting positions of the TPP resonance wavelength in the reflection spectrum collected for both before and after antigen binding in the PSi TPP biosensor, and then calculating the shift of the TPP resonance wavelength. More biomolecule binding in the PSi TPP biosensor will cause more redshift. This is the mechanism of the quantitative detection of biomolecules. Alternatively, there is also a machine learning (ML) approach for the qualitative determination of the detection results for the signal and data processing of the PSi TPP biosensor [35]. Moreover, the PSi TPP biosensor is capable of the real-time detection of biomolecules. This is based on recording the reflection spectrum in real time as the biomolecules bind on the biosensor surface. We demonstrated this capability by detecting the Nucleocapsid protein of SARS-CoV-2 in real time in previous publications [27,28].

As shown in the picture on the left of Figure 1, for this work, we used an in-house developed optical spectral measurement system, which is a high throughput system with signal reading in accordance with a sensor array format of 12 × N. It mainly consists of a two-dimensional moving stage to carry and position the sensor array and a fiber spectrometer to take the optical measurement. It takes one second to read the reflection spectrum of each biosensor and moves the fiber probe to the next biosensor. Thus, it takes 96 s to complete the sequential reading of the optical signals of the maximum 12 × 8 sensor array.

## 3. Results and Discussion

Figure 2a shows the cross-sectional scanning electron microscopy (SEM) image of the fabricated PSi TPP device. The periodic structure of PSi DBR and the thin gold film on top of DBR are clearly visible. Figure 2b shows the atomic force microscopy (AFM) image of the Au thin film surface morphology, which clearly demonstrates the porous structure of Au thin film. The porous nature of the gold thin film is due to the conformal deposition of gold on porous silicon. Figure 2c shows the COMSOL V5.5 simulated electrical field distribution pattern at the resonance wavelength of the TPP resonant mode. It can be observed from the colormap that the peak of the electrical field resides in the first PSi layer. This PSi layer is also a nanocomposite layer wherein Au nanoparticles generated during the sputtering process can infiltrate porous silicon pores and immobilize on the pore walls [31]. In turn, CagA antigen molecules can also infiltrate into PSi pores and bind with antibodies, specifically inside the pores of the first PSi layer. This ensures the spatial overlap between antigen–antibody binding and the peak of the TPP modal field. Figure 2d shows an example of the measured reflection spectrum of the PSi TPP device. The resonance dip shows TPP resonance where light energy at the resonant wavelength is coupled into the TPP resonant mode and strong electrical field confinement appears in the proximity to the Au–PSi interface. Figure 2e shows an example of the redshift of TPP resonance upon the specific binding of the CagA antigen with antibodies immobilized on the PSi TPP biosensor beforehand. The binding of antigens interacts with locally confined electrical field of TPP resonance. As a result of this perturbation of the TPP mode, the TPP resonance wavelength shifts towards longer wavelengths. Figure 2f shows that there appears to be no redshift when the antibody biofunctionalized PSi TPP biosensor is exposed to PBS buffer.

To characterize the response of the PSi TPP biosensor, different concentrations of CagA antigen were detected with the PSi TPP biosensor. The redshift vs. antigen concentration can be plotted and the data can be fitted with the linear trend. Figure 3 shows the response curve of the PSi TPP biosensor for detecting the varying concentrations of the CagA antigen. At each CagA concentration, five detection experiments were conducted with each experiment using a different biosensor. As a result, a total of sixty biosensors were used to obtain the data in Figure 3. A high-quality linear fitting is also given to match the data points at different concentrations. The sensitivity of the PSi TPP biosensor can be defined as:(2)$$S_{TPP}=\frac{d\lambda_{Res}}{dC}$$
where $S_{TPP}$ is the sensitivity of the PSi TPP biosensor in pm/(ng/mL), $\lambda_{Res}$ is the TPP resonance wavelength in nm, and C is the CagA antigen concentration in ng/mL. The response curve of the PSi TPP biosensor, which is the redshift of the TPP resonance wavelength vs. CagA antigen concentration, can be considered a linear response. The deviation from the linear trend at a high concentration is due to the saturation effect of the biosensor, that is, the red shift increment tends to be less obvious at higher CagA concentrations. This is a common effect of biosensors due to the limited available binding sites provided by immobilized antibodies. In practical applications, only the linear range of the biosensor is used for quantitative detections.

From the linear fitting of the experimental data obtained at different antigen concentrations, the sensitivity, which is the slope of the fitted curve according to Equation (2), is 100 pm/(ng/mL). Given the sensitivity, the limit of detection can be calculated to be:(3)$$LOD=K\frac{SD}{S_{TPP}}=0.05\;ng/mL$$
where K is a factor of 3, SD is the standard deviation of response signal level of the blank sample (here PBS is used as shown in Figure 4, and $S_{TPP}$ is the sensitivity. The linear range of the biosensor for the CagA antigen detection is 0.25–8 ng/mL, which coincides with the range of CagA antigen concentration in clinical stool specimens [38].

For control experiments, we used the PSi DBR biosensor consisting of PSi DBR with 5 nm titanium thin film and 10 nm gold thin film consecutively deposited atop. The 10 nm gold thin film is not enough to support TPP and only serves as a platform for antibody immobilization. The response of the PSi DBR biosensor is based on the redshift of the central wavelength of the PSi DBR photonic bandgap. The results of detecting the varying concentrations of the CagA antigen is also shown in Figure 3. It can be seen that the sensitivity of the PSi TPP biosensor is more than 15 times that of the PSi DBR biosensor. This is due to the resonance mode of TPP that can have a better field confinement than DBR. DBR only has a photonic bandgap but no resonance mode.

For the specificity test of the PSi TPP biosensor, BabA antigen, VacA antigen, BSA (bovine serum albumin), uric acid, PBS buffer, as well as glucose, were used as nonspecific targets of detection. For the introduction of interference, all the nonspecific species have concentrations of 5 ng/mL, and the CagA antigen has a concentration of 1 ng/mL. Figure 4 shows the comparison of the specific response signal (the redshift for CagA antigen) with the nonspecific response signal (the redshift for other targets). The specific response signal is at least six times stronger than the nonspecific response signal in terms of redshift amplitude. This demonstrates the good specificity of the PSi TPP biosensor in the aim of detecting the CagA antigen of *H. pylori*. In addition, to further determine the specific recognition of *H. pylori*, a competitive test was also carried out wherein two specimens consisting of 1 ng/mL CagA mixed with 5 ng/mL BabA and 5 ng/mL VacA, respectively, were used for detection. The response signal level of the competitive test is also included in Figure 4 for comparison. It can be observed that, under high dose interference, the PSi TPP biosensor can successfully recognize the CagA antigen.

To compare our proposed PSi TPP biosensor with biosensors reported in the literature, Table 4 shows a comparison of the performances of the PSi TPP biosensor with other state-of-the-art biosensors for the detection of the CagA antigen/antibody. This demonstrates that the PSi TPP biosensor has a superior performance and is competitive for application in the onsite rapid diagnosis of *H. pylori* infection. In addition, due to the compatibility of porous silicon fabrication with CMOS processes, the integration of the PSi TPP biosensor with silicon-based optoelectronics and electronics is of significant interest. Silicon-based optoelectronics can offer a light source, filters, couplers, and detectors. Si-based electronic integrated circuits can implement the spectral data-processing algorithm. Therefore, an optical measurement system can be implemented as an integrated circuit chip. Furthermore, the PSi TPP biosensor can be integrated together with an optical measurement system in the same package or even on the same die. This is a great advantage for the PSi TPP biosensor compared with the other biosensors. First, CMOS-compatible integration can scale up the fabrication volume and the lower fabrication cost of both the biosensors and measurement system due to the capabilities of the existing CMOS infrastructure. Second, integration can reduce the form factor of the measurement system so that ubiquitous handheld detection can be realized. Third, integration can improve the reliability and repeatability of measurement results since process variations are minimized. These factors, when combined, will ultimately promote the practical applications of the PSi TPP biosensor in clinical settings.

## 4. Conclusions

In this original research work, we have developed and demonstrated for the first time, to the best of our knowledge, a high-performance porous silicon TPP optical biosensor for the detection of the CagA antigen of *H. pylori* bacterium, with an application in the diagnosis of *H. pylori* infection. The PSi TPP biosensor is composed of a porous silicon multilayer structure, which is a DBR, and a plasmonic layer, which is gold thin film. These two components thereafter comprise the PSi TPP device and the optical biosensor. After optimization in the biosensor design, fabrication, and biofunctionalization, the sensitivity of the PSi TPP biosensor can reach up to 100 pm/(ng/mL), and the limit of detection can be as good as 0.05 ng/mL. In addition, the good specificity of the PSi TPP biosensor is demonstrated by the positive-to-negative ratio of a factor greater than six. These experimental results demonstrate the characteristics of the PSi TPP biosensor for CagA antigen detection and therefore its suitability for *H. pylori* infection diagnosis.

There are a lot of future research opportunities concerning the PSi TPP biosensor. Researchers can carry out the further optimization of the PSi TPP biosensor by incorporating tunable epsilon-near-zero material, such as graphene, into the TPP device structure, to replace noble metal material [41]. Graphene has the property of tunable conductivity, thus allowing the tuning of the TPP resonance wavelength and the electrical field confinement profile of the TPP resonant mode. Also, optical Tamm state (OTS) [42], which can be confined at the interface between two symmetric or asymmetric photonic crystals, can couple with the TPP resonant mode, and the resulting Fano resonance [43] can have a tunable field confinement profile depending on the coupling strength between OTS and TPP modes.

In summary, an optical biosensor based on PSi TPP is an attractive biosensing technology for detecting pathogens and their related antigens. It has vast potential for the rapid onsite detection required in scenarios such as population screening, epidemic surveillance, and personal healthcare. Well poised to act as an onsite rapid diagnostic tool for POCT applications, PSi TPP biosensor can serve as an indispensable complement to central analytical lab-based detection techniques such as PCR and sequencing. Therefore, the PSi TPP biosensor can be an efficient platform to curb pathogen dissemination and help improve healthcare outcomes.

## Figures and Tables

**Figure 1 sensors-24-05153-f001:**
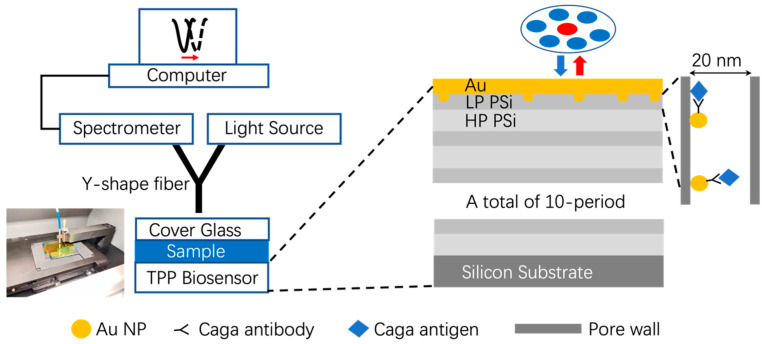
The PSi TPP biosensor structure and its optical measurement configuration. The Au NPs embedded in the first LP PSi layer means that Au NP can infiltrate into the nanopores of porous silicon. The picture on the bottom left shows the setup of the 12 × 8 biosensor array and its sequential measurement by an in-house developed equipment mainly consisting of a two-dimensional moving stage to carry and position the sensor array and a fiber spectrometer to take the optical measurement.

**Figure 2 sensors-24-05153-f002:**
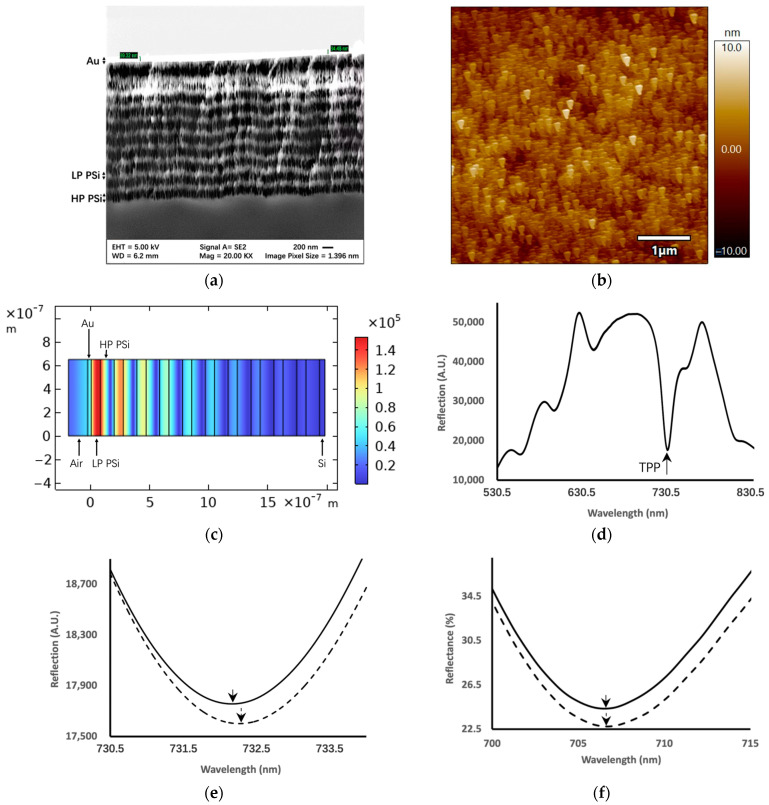
(**a**) Cross-sectional scanning electron microscopy image of the TPP biosensor structure. The periodic layered structure of the porous silicon DBR is clearly visible. The gold thin film is the bright layer on top of the porous silicon DBR with the thickness not to scale due to a focusing issue; (**b**) atomic force microscopy image of a Au thin film surface morphology, showing the nanoporous structure of Au due to conformal deposition onto porous silicon; (**c**) Color map of electrical field strength distribution profile of TPP biosensor simulated by COMSOL V5.5. The field peak resides in the first (or top) PSi layer close to the thin metal film. Light is incident from the left side of the TPP device in air with a power of 1 W/m; (**d**) An example of the reflection spectrum of the TPP device is that of where the resonant state manifests as the reflection minimum at a wavelength of around 730 nm; (**e**) Example of the redshift of the TPP resonance wavelength upon the specific biomolecular binding of 3 ng/mL CagA antigen with the CagA antibody immobilized on the biosensor surface beforehand. The spectrum of the TPP biosensor both before (solid curve) and after (dashed curve) binding is shown. The redshift of the resonance minimum, indicated as arrows, is around 360 pm. (**f**) Example TPP resonance spectra upon exposure to PBS buffer with CagA antibody immobilized on the biosensor surface beforehand. The spectrum of the TPP biosensor both before (solid curve) and after (dashed curve) PBS is shown. The shift of resonance minimum, indicated as arrows, is 0 pm.

**Figure 3 sensors-24-05153-f003:**
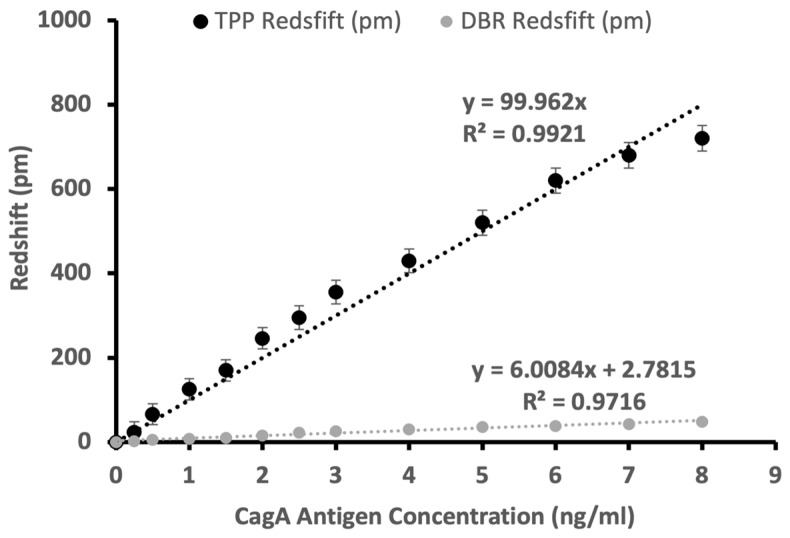
Response characterization of CagA antigen detection for detecting varying concentrations of CagA antigen in the PBS buffer, with both PSi TPP (black circles) and PSi DBR (grey circles) biosensors. Error bars on the experimental data points (solid circle) show a standard error from five experiments, with each experiment using a different biosensor. The linear fittings (dashed curves) are performed to match the data points. The linear equations go through the origin and the quality of the fitting is also given in the figure.

**Figure 4 sensors-24-05153-f004:**
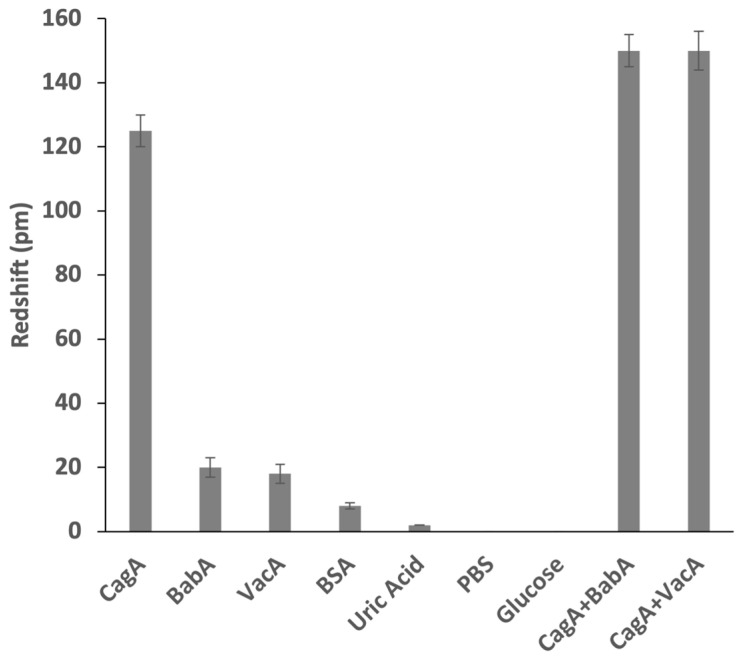
Specificity test and competitivity test of the PSi TPP biosensor with the objective of CagA antigen detection. CagA has a concentration of 1 ng/mL and all other nonspecific species concentrations of 5 ng/mL. Error bars on the experimental data bars (solid rectangle) show the standard error from five experiments, with each experiment using a different biosensor.

**Table 1 sensors-24-05153-t001:** Comparison of different *H. pylori* detection techniques.

Detection Techniques		Pros	Cons
Non-invasive test methods (endoscopy not required)	Serological test [6]	No side effects on patients	Positive results may not mean persistent infection
	Stool antigen test [5]	Easy operation	Fecal sample handling may affect results
	Urea breath test [4]	Easy operation	Drugs taken by patients may impact accuracy
	Biosensors [7,8,9]	Easy operation, high sensitivity	Specificity may be affected by interferents in samples
Invasive test methods (endoscopy required)	Endoscopy [10]	Gastric pathology observation available	Accuracy may vary drastically depending on operator experiences
	Culturing [12]	Highly specific due to controlled culturing conditions	Long turnaround time and vulnerability of results to culturing conditions
	Histological test [11]	High sensitivity and specificity	Accuracy may vary drastically depending on operator experiences
	Rapid urease test [13]	Rapid and easy to operate	Bacterium may not be present in biopsy
	Polymerase chain reaction (PCR) [14]	Accurate and high throughput capability	Need clean environment, samples may contain PCR blockers

**Table 2 sensors-24-05153-t002:** Porous silicon electrochemical anodization conditions and optical parameters of the resulting porous silicon material.

Porous Silicon Layer	Current Density	Anodization Time	Porosity of Layer	Optical Refractive Index	Thickness of Layer
Low porosity (LP PSi)	5 mA/cm^2^	20 s	52%	2.08	100 nm
High porosity (HP PSi)	48 mA/cm^2^	6 s	76%	1.46	150 nm

**Table 3 sensors-24-05153-t003:** Biofunctionalization procedure of the PSi TPP biosensor for CagA detection.

No.	Step Name	Step Operation
1	Chip cleaning	Clean the surface of the chip separately with anhydrous ethanol and ultrapure water, blow dry with N_2_, and set aside.
2	Carboxylation modification	Dilute the polycarboxylation reagent (Xlement Cat. No. G40005) to 100 µM working concentration with ultrapure water, take an appropriate amount of carboxylation reagent and add it to the surface of the chip. Leave it at 4 °C overnight or 37 °C for 4 h.
3	Activation of functional groups on surface of biosensors	Wash the carboxylated modified chip three times with ultrapure water and blow dry with N_2_. Prepare the experimental chip card following standard operating procedures and connect it to the corresponding flow interface in sequence. Prepare an activation reagent solution with a final concentration of 10 mM using 100 mM activation buffer solution (Xlement Cat. No. S20028), and inject 100 µL activation reagent (prepare when need to use) into the chip with a flow rate of 10 μL/min. Running buffer is the activation buffer.
4	Preparation of immobilization antibody solution	Prepare a solution of capture antibodies against CagA antigen of *Helicobacter pylori* and dilute with a coupling buffer solution (Xlement Cat. No. S20029) to 15 μg/mL concentration for later use.
5	Chip conjugation with antibody	Take an appropriate amount of capture antibody (CagA antibody) solution and add it to the surface of the chip. Inject 100 μL of the capture antibody solution into the chip at a flow rate of 10 μL/min. Running buffer is the conjugation buffer.
6	Sealing	Take an appropriate amount of sealing buffer solution (Xlement Cat. No. G30004) and add it to the surface of the chip. Inject 100 μL sealing buffer solution into the chip with a flow rate of 10 μL/min. The running buffer is the sealing buffer.
7	Storage	The biosensors can be stored for 3–5 days under 37 °C, two months under 25 °C, and 6 months under 2–8 °C.
8	Biosensing	Restore the biosensors to a room temperature of 25–27 °C. Drop 20 µL of CagA antigen solution in varying concentrations in PBS (phosphate-buffered saline, pH 7.4) buffer on biosensor surface. Put on cover glass and take spectral measurement with fiber spectrometer.

**Table 4 sensors-24-05153-t004:** Comparison of the biosensing performances of CagA antigen/antibody detection for *H. pylori* diagnosis.

Biosensor Technique	Sensitivity	LOD (ng/mL)	Linear Range (ng/mL)	Positive-to-Negative Ratio
Amperometric [38]	656.0 µA/(ng/mL)	0.10	0.1–8	>5
Electrochemical [39]	0.275 µA/(ng/mL)	0.05	0.05–50	~4
Voltametric [40]	1.0 µA/(ng/mL)	0.20	0.2–50	>5
TPP (This work)	100 pm/(ng/mL)	0.05	0.25–8	>6

## Data Availability

The datasets generated and/or analyzed during the current study are not publicly available due to privacy but are available from the corresponding author upon reasonable request.

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
