# Peer review of "Optical Biosensor Based on Porous Silicon and Tamm Plasmon Polariton for Detection of CagA Antigen of Helicobacter pylori"

_sensors, 2024, doi:10.3390/s24165153_

Round 1

Reviewer 1 Report

Comments and Suggestions for Authors

The paper introduced a novel optical biosensor leveraging the combination of porous silicon and TPP, demonstrating innovation in biosensing technology. And it exemplified the potential of integrating nanotechnology with photonics, offering a new perspective in the field of biosensors. The reported high sensitivity (100 pm/(ng/ml)) and good specificity of the biosensor are crucial for enhancing the accuracy of disease diagnostics. The potential use of the sensor in POCT applications offers new possibilities for rapid on-site diagnostics. In conclusion, this study showcased significant innovation and application potential in the field of biosensor technology. However, to enhance its scientific rigor and practical application value, the authors are advised to conduct in-depth and more research work to overcome the weaknesses.

1. The authors did not provide information on the sensor's long-term stability and reusability, necessary for assessing its reliability in practical applications.

2. The research did not elaborate on how environmental factors, such as temperature, pH value, might impact the sensor's performance, which could affect stability and accuracy.

3. The study lacked data from tests conducted on real clinical samples, which was essential for validating the sensor's performance in actual clinical settings.

4. The paper discussed TPP, which is a phenomenon similar to surface plasmon polaritons, involving electromagnetic surface states at the interface of metal and dielectric. TPP can be considered a special type of Surface Plasmon Polariton. In the Introduction, the author should briefly describe the differences between their TPP sensor and other SPR/SPP sensors (Such as: https://doi.org/10.1002/adfm.202314481, https://doi.org/10.1002/smll.201906108,  https://doi.org/10.3390/s21093191).

I hope these suggestions will be helpful to the authors, and I look forward to seeing these problems addressed in reviewed paper.

Comments on the Quality of English Language

The grammatical structure appears to be correct, and the syntax is used effectively to convey complex scientific concepts.

Reviewer 2 Report

Comments and Suggestions for Authors

The manuscript entitled "Optical Biosensor based on Porous Silicon and Tamm Plasmon Polariton for Detection of CagA Antigen of Helicobacter pylori" demonstrates a porous silicon-based TPP optical biosensor for the detection of the CagA antigen for diagnosing H. pylori infection. This paper is original and shows promising performance of the PSi TPP optical biosensor with a sensitivity of up to 100 pm/(ng/ml), a LOD of 0.05, and specificity for CagA detection. The novelty of this paper is commendable. However, there are some questions and concerns that need to be addressed before publishing this paper in Sensors. Here are some comments:

1. In biosensor experiments, it is important to repeat the molecular detection experiments with different devices or provide statistics of the biomolecular signal using batch-fabricated devices. Many factors and mechanisms involved in the sensing experiment can lead to misleading results, such as fouling issues. In this paper, the authors provide one device or a very limited number of devices and do not show the statistics of the CagA sensing results.

2. And in Figure 3, it is unclear if these CagA concentration-dependent detection experiments were conducted with the same optical sensor device or different devices..  

3. Are these optical biosensors intended for one-time use? If not, is there a protocol for proper cleaning/activation before reuse?

4. If they are for one-time use, what is the sensor density per chip? I assume there are arrays of TSi TPP optical biosensor devices on one chip, and during each CagA sensing experiment, every device will be exposed to CagA. In this case, detecting more than one optical biosensor CagA signal (redshift) in a single detection becomes possible (as mentioned in question 1). The read throughput and sensor density are important. The authors mention the CMOS-compatible process for making these TPP PSi optical biosensors; therefore, the density of sensors should be discussed. 

5. A small suggestion for Figure 1: The layer stack of the TPP biosensor can be presented better. It appears that the Au layer is on top of the stacked PSi layer, but showing that this Au layer infiltrates inside the pores of PSi is also important. Inserting some Au color into the first layer of LP PSi can improve the presentation.

6. In the conclusion section, the second paragraph starting with "There are a lot..." is too long. After all, this is not a review paper. The authors should consider trimming this paragraph. A brief perspective on further development will suffice.

Comments on the Quality of English Language

The quality of English is decent. The issues within language part, is there are a lot repeated abbreviation and its original meaning, for example, Tamm Plasmon Polarition (TPP) and cytotoxin-associated antigen A (CagA) and etc., These abbreviation should appear one time in the main text. 

Reviewer 3 Report

Comments and Suggestions for Authors

Dear Authors,

The paper is written concisely and well.

My suggestion is that the conclusion is too long.

Also in the conclusion part is where compatibility is written of porous silicon fabrication with CMOS processes, integration of PSi TPP biosensor with silicon-based optoelectronics and electronics explained advantage and importance that part should be reduced. Also, transfer that same part in the results and discussion section, when that part is transferred in the discussion, it is necessary to write the same, expand and explain all those individual parts of the importance of production and use.

The paper lacks part of the clinical application. Did you have the opportunity to conduct more intensive trials on a larger number of patients? Or is it your opinion that it was not necessary.

Round 2

Reviewer 1 Report

Comments and Suggestions for Authors

The authors have answered all the questions.

Reviewer 2 Report

Comments and Suggestions for Authors

I have reviewed the revised manuscript and am pleased to report that the authors have adequately addressed my previous questions and concerns. The revisions enhance the quality and clarity of the paper, and I now support its publication in Sensors. The authors' responses were thorough and satisfactory, and the manuscript now meets the necessary standards for publication.